# High-Resolution Crystal Structure of Muscle Phosphoglycerate Mutase Provides Insight into Its Nuclear Import and Role

**DOI:** 10.3390/ijms232113198

**Published:** 2022-10-30

**Authors:** Janusz Wiśniewski, Jakub Barciszewski, Jakub Turlik, Karolina Baran, Przemysław Duda, Mariusz Jaskolski, Dariusz Rakus

**Affiliations:** 1Department of Molecular Physiology and Neurobiology, Faculty of Biological Sciences, University of Wrocław, 50-335 Wrocław, Poland; 2Institute of Bioorganic Chemistry, Polish Academy of Sciences, 61-704 Poznań, Poland; 3Department of Protein Engineering, Faculty of Biotechnology, University of Wrocław, 50-383 Wrocław, Poland; 4Department of Crystallography, Faculty of Chemistry, Adam Mickiewicz University, 61-614 Poznań, Poland

**Keywords:** phosphoglycerate mutase, protein structure, 14-3-3, nucleolus, protein–protein interaction, mTOR

## Abstract

Phosphoglycerate mutase (PGAM) is a glycolytic enzyme converting 3-phosphoglycerate to 2-phosphoglycerate, which in mammalian cells is expressed in two isoforms: brain (PGAM1) and muscle (PGAM2). Recently, it was shown that besides its enzymatic function, PGAM2 can be imported to the cell nucleus where it co-localizes with the nucleoli. It was suggested that it functions there to stabilize the nucleolar structure, maintain mRNA expression, and assist in the assembly of new pre-ribosomal subunits. However, the precise mechanism by which the protein translocates to the nucleus is unknown. In this study, we present the first crystal structure of PGAM2, identify the residues involved in the nuclear localization of the protein and propose that PGAM contains a “quaternary nuclear localization sequence (NLS)”, i.e., one that consists of residues from different protein chains. Additionally, we identify potential interaction partners for PGAM2 in the nucleoli and demonstrate that 14-3-3ζ/δ is indeed an interaction partner of PGAM2 in the nucleus. We also present evidence that the insulin/IGF1–PI3K–Akt–mTOR signaling pathway is responsible for the nuclear localization of PGAM2.

## 1. Introduction

Phosphoglycerate mutase (PGAM, EC 5.4.2.1) is an evolutionarily conserved enzyme that catalyzes the conversion of 3-phosphoglycerate (3-PG) to 2-phosphoglycerate (2-PG) as the eighth step of glycolysis. Mammalian PGAM is encoded and expressed as two isoforms: the brain (PGAM1, also called PGAM-B) and the muscle (PGAM2, PGAM-M) isoform. PGAM1 is expressed more abundantly than PGAM2 in all tissues except skeletal muscles, where PGAM2 dominates. The enzyme functions as a homodimer, which can be composed from the brain-only (PGAM1–PGAM1) or muscle-only (PGAM2–PGAM2) subunits, or as a heterodimer (PGAM1–PGAM2) [1].

PGAM1 has been studied by X-ray crystallography and numerous structures, determined in the absence and presence of the 3-PG substrate [2,3,4], were deposited in the Protein Data Bank (PDB). In contrast, no PGAM2 structure has been solved using experimental, crystallographic or NMR, data and the only available model was generated by theoretical calculations, using the AlphaFold software [5,6].

Numerous conserved enzymes engaged in carbohydrate metabolism also display non-enzymatic functions, and can thus be classified as moonlighting proteins. Besides their cytoplasmic localization, the glycolytic enzymes are present in the nucleus [7] and mitochondria, where their role is not limited to catalysis [8]. Hexokinase, for instance, was found to bind to the outer mitochondrial membrane and to regulate its permeability, thus inhibiting the apoptosis process [9]. The muscle isoform of phosphofructokinase is known to interact with neuronal nitric oxide synthase [10], whereas aldolase regulates the assembly of the actin cytoskeleton and the stability of the neurofilament light chain transcript [11,12]. Similarly to aldolase, enolase was characterized as a factor regulating axon growth and the activity of RHOA kinase [13]. It has been also observed that PGAM, independently of the concentration of the glucose metabolites, is directed to nucleoli upon the stimulation of the insulin/insulin-like growth factor 1 (IGF1)-phosphoinosidite 3-kinase (PI3K) [14,15]. Nucleolar PGAM stabilizes the nucleoli structure, ensures mRNA expression and, presumably, is a prerequisite for the assembly of new pre-ribosomal subunits [10]. It has been also demonstrated that only the PGAM2 isoform is essential for the nucleolar localization of the PGAM homo- and heterodimers [15].

Nuclear translocation of proteins of the mass higher than about 50 kDa requires a complex formation between nuclear localization sequence (NLS)-containing cargo and importins. However, it is well-known that members of the 14-3-3 protein family also affect the nuclear retention of some proteins. 14-3-3s interacting with phosphorylated serine/threonine residues can mask NLS or nuclear export signal (NES) within its binding partner structure. It can also block the binding of other interactors and induce conformational changes within the target protein [16]. Some of the 14-3-3s’ binding partners play critical functions within the cell nucleus, e.g., the mitotic phosphatase CDC25, the histone deacetylase HDAC4, and the transcription coactivators YAP and TAZ [17].

In this study, we present the first crystal structure of the muscle isoenzyme of PGAM, solved using X-ray diffraction data extending to 1.29 Å resolution. The structure enabled us to predict the residues involved in nuclear localization, and we verified their functionality using site-directed mutagenesis. Moreover, we demonstrate that the 14-3-3ζ/δ protein is the binding partner of PGAM2 in the cell nucleus, and we provide evidence that the insulin/IGF1-PI3K-protein kinase B (AKT)-mammalian target of rapamycin (mTOR) signaling pathway is responsible for the regulation of nuclear localization of PGAM2.

## 2. Results and Discussion

### 2.1. Crystal Structure of PGAM2

We present the crystal structure of rabbit muscle phosphoglycerate mutase PGAM2 determined at 1.29 Å resolution. This is the first structure of the muscle isoenzyme of PGAM. PGAM2 crystallized in the P41212 space group, with one protein chain in the asymmetric unit (Figure 1A). The quality of the electron density is excellent (Figure 1D), except for the C-terminal residues 245–253 which are disordered and were, therefore, not modeled. Details of data collection and refinement are presented in Table 1. 

The biological assembly of PGAM2, as predicted by PISA assembly [18] and based on homology with PGAM1 structures (PDB codes: 1YFK, 4GPI, 4GPZ, [2,4]), is a homodimer, with the second subunit generated by the crystallographic twofold axis. The two PGAM2 subunits interact via a pair of salt bridges between the side chains of residues R65 and D72, an ionic interaction of residues R83 interestingly mediated by a chloride ion, and by π–π stacking of the indole rings of residues W68. These interactions are located in a central hydrophobic region formed by antiparallel α-helices 60–74, loops 75–80 and short antiparallel β-strands 81–83 (Figure 1B). At the edge of the dimer interface, two additional sets of interactions are formed by the R140 residues of one subunit and F52, D53, and Q76 of the complementary subunit (Figure 1C). The fact that identical interactions are found at the interface of PGAM1 homodimers [2,4] provides a convincing and elegant explanation for the existence of PGAM1–PGAM2 heterodimers. 

Comparison of our model with the AlphaFold [5,6] prediction for human PGAM2 shows that the models generally agree (RMSD of 1.53 Å for 243 superimposed Cα atoms), except for the last 18 C-terminal residues. In the prediction they form a helix covering the active site of the enzyme, while in our model they are not visible in the electron density, thus most likely disordered (Figure 2A). It must be noted that the differing region has the lowest confidence score of the whole prediction structure.

The overall fold of PGAM2 is similar to PGAM1, as can be expected for proteins with over 80% sequence identity (Appendix A). Superposition of the present PGAM2 model with PGAM1 model 4GPZ (highest resolution PGAM1 structure in the PDB) gives an RMSD of only 0.729 Å for 244 superimposed Cα atoms. There are, however, a number of differences between the isoforms which are interesting in the context of both metabolic and non-metabolic functions of PGAM2. It was shown for PGAM1 that the negatively charged side chain of glutamate residue E19 is located in the active site and can interfere with substrate binding and phosphorylation of histidine residue H11, which is necessary for catalysis. In the active, H11-phosphorylated form of PGAM1, a conformational change positions E19 away from the active site thus removing the interference. Phosphorylation of tyrosine residue Y26 promotes this conformational change, as shown by kinetic and mutational studies. It must be noted, however, that direct structural evidence for this is lacking and no Y26-phosphorylated PGAM1 structure is available [2]. PGAM2 position 26 is occupied by a phenylalanine residue instead of a tyrosine residue and thus this isoenzyme is unable to undergo such a phosphorylation. Examination of the PGAM2 structure presented here shows that residue E19 is pointed away from the active site, as in the active H11-phosphorylated PGAM1 form (Figure 2B). This might mean that PGAM2 is constitutively more active than PGAM1. However, the verification of this hypothesis and its potential cellular implications lie outside of the scope of this paper.

Another important difference between PGAM2 and PGAM1 is the presence in PGAM2 of surface-exposed, positively charged residues, conserved in mammalian PGAM2, but not PGAM1 – K33, K49, K129, K146 (Figure 3A, Appendix A). Although PGAM was shown to enter the cell nucleus and this translocation to be reliant on the presence of at least one PGAM2 molecule in the PGAM dimer [15], until now the NLS remained unidentified. Analysis with NLS mapper [19] did not suggest any candidates for the putative canonical or bi-partite NLS with above-threshold score. Using our structural data, we selected the above-mentioned lysine residues as candidates for the building blocks of the elusive non-canonical NLS of PGAM2 and performed mutagenesis studies. K33 was not included in this experiment, as it is located in the immediate vicinity of the active site (Figure 3A).

### 2.2. Identification of PGAM2 NLS

We designed and purified a series of recombinant human PGAM2 forms, each carrying one of the following mutations: K49G, K129D and K146T. The substitutions introduced into the PGAM2 sequence were the residues present at equivalent positions of human PGAM1 (Figure 4). We then analyzed the nuclear import of PGAM2 by transfecting KLN-205 cancer cells with fluorescently labeled PGAM2 mutants (Figure 4). No change in nuclear fluorescence was observed for the K49G and K129D variants. Mutation K146T, however, decreased the nuclear fluorescence intensity by about 30% (Figure 4). Specifically, the mean nuclear fluorescence intensity (arbitrary units) was 7.98 ± 2.61 in WT-transfected cells, 7.80 ± 2.33 in K49G-transfected cells, 7.58 ± 2.02 in K129D-trasnfected cells, and 5.73 ± 1.45 in K146T-transfected cells (F = 10.42, *p* = 0.000002, one-way ANOVA). We thus conclude that K146 is a part of the PGAM2 NLS, although likely not the only one, as nuclear localization was not completely abolished by K146 mutation. 

A closer look at the position of K146 in the PGAM2 sequence reveals that it is located in the region 138KERRYAGL**K**-(X)29-KAGKR180 (K146 in bold). In the following, the peptide 138–146 (KERRYAGLK) will be referred to as part A and 176–180 (KAGKR) as part B of the proposed NLS of PGAM2. NLS motifs similar to part A, containing 3 to 5 basic residues separated with short spacers, such as KRAAER [21] or RKVNKRLK [22], were previously identified as bi-partite NLSs with unusually long spacers of 18 [23] or 24 residues [24]. Examination of the K146 position in the 3D structure of the PGAM2 subunit reveals that part B is situated on the other side of the protein molecule, and is thus unlikely to participate in the same interactions as part A. However, in the PGAM2 dimer, each part A lies in close proximity of part B from the complementary subunit, the residues K138 (A), R140 (A), R141 (A), K176 (B), K179 (B), and R180 (B) forming two almost contiguous stretches of surface-exposed basic residues spanning around 32 Å each. The critical residue K146 is somewhat removed from this stretch, with 13.2 Å Cα-Cα distance to the nearest other basic residue (R140), but still close enough to participate in the same protein–protein interactions (Figure 3B). We hypothesize, therefore, that together these residues may form a structural “quaternary NLS”. Additionally, PGAM1 and PGAM2 share a patch of positive potential on the surface near part B of the hypothetical NLS (Figure 3A), while residues K176, K179 and R180 are conserved in PGAM1 and PGAM2. This may explain why not only PGAM2 homodimers but also PGAM1–PGAM2 heterodimers can be successfully imported into the cell nucleus.

### 2.3. Identification of Nucleolar Binding Partners of PGAM2

Our previous studies showed that in nucleolar protein extracts, after chemical cross-linking, PGAM2 forms complexes with numerous ribosomal and family 14-3-3 proteins [15]. To confirm these interactions, we performed in the present study affinity chromatography experiments on nucleolar protein extracts, using PGAM2 immobilized on Sepharose resin, without any previous cross-linking reaction. The affinity chromatography step was followed by protein identification using LC-MS/MS mass spectrometry. The proteins identified at the confidence level above 95% (MASCOT score > 90 [25,26]) are listed in Appendix A. From this list, 14-3-3ζ/δ and 60S acidic ribosomal protein P0 (RPLP0) were selected for further investigations, as representatives of both protein groups, scoring high in both the present and previous analysis. It must be noted, however, that a number of other additional potential partners of PGAM2 were identified in the present experiment, including nucleolin and lamin A/C. 

Since the 14-3-3ζ/δ–PGAM2 interaction was detected using extracts of nucleolar proteins, we next conducted a co-localization experiment in the KLN-205 cells. We observed that the 14-3-3ζ/δ- and PGAM2 C-end-related fluorescence signals co-localize within the nucleolar regions (Figure 5A).

To verify the formation of PGAM2 complexes with 14-3-3ζ/δ and RPLP0 in the cell, we employed the proximity ligation assay. We detected fluorescence signals related to the 14-3-3ζ/δ–PGAM2 interaction not only in cell nuclei, but also in the cytoplasm (Figure 5B). However, we did not observe any signal associated with the RPLP0–PGAM2 complex. This may indicate the absence of the PRLP0–PGAM2 interaction in the cell, but it may also reflect the shortcomings of the method we used; for instance, the distance between the antibodies interacting with the binding partners might be too long for hybridization of the nucleic acids associated with the antibodies.

To obtain a deeper insight into the mechanism regulating the 14-3-3ζ/δ–PGAM2 complex formation, we cultured KLN-205 cells in conditions that were previously described to block the nuclear localization of PGAM2 [15]. Culturing the cells for 2 h in serum-depleted medium or in the presence of 1 μM wortmannin (an inhibitor of PI3K), showed a decreased amount of PGAM2 localized in the nuclei (Figure 6A). Specifically, the mean nuclear-to-cytoplasmic fluorescence intensity ratio of about 1.28 ± 0.3 was recorded in control conditions, 0.77 ± 0.19 in cells cultured in serum-free medium, and 0.9 ± 0.14 in presence of wortmannin (F = 42.65, *p* = 1 × 10^−13^, one-way ANOVA). The result was accompanied by both a lowered 14-3-3ζ/δ nuclear-to-cytoplasmic fluorescence intensity ratio (2.58 ± 1.53 in control conditions, 1.41 ± 0.47 in cells cultured in serum-free medium, and 1.57 ± 0.53 in presence of wortmannin, F = 12.73, *p* = 0.00001, one-way ANOVA) (Figure 6B) and 14-3-3ζ/δ–PGAM2 interaction (mean fluorescence intensity in the nuclear regions of about 18.28 ± 3.64 in control condition, 8.69 ± 2.25 in cells cultured in serum-free medium, and 7.19 ± 1.92 in presence of wortmannin, F = 147.55, *p* = 1 × 10^−16^, one-way ANOVA) (Figure 6C). The 14-3-3ζ/δ–PGAM2 interaction is strongly correlated with the nuclear localization of PGAM2, since inactivation or inhibition of the insulin/IGF1-PI3K signaling pathway abolished the binding of these proteins (Figure 6C). 

Members of the large 14-3-3 protein family are abundantly expressed in all eukaryotic cells [26]. In mammals, seven 14-3-3 proteins encoded by seven different genes have been described, namely α/β, ε, γ, σ, ζ/δ, τ/θ, and η [27]. They are engaged in regulation of a wide spectrum of cellular processes, such as cell proliferation, enzymatic activity, protein–protein interactions, proteolysis and phosphorylation status of target proteins, autophagy, and protein subcellular localization [28]. Moreover, disfunction of several 14-3-3 binding partners is associated with cancer, diabetes, and neurological disorders [29]. 14-3-3 proteins modulate the activity of target proteins through binding to phosphoserine- or phosphothreonine-containing motifs [30]. However, non-phosphorylated binding motifs have also been identified [31,32]. All 14-3-3 isoforms recognize two high-affinity binding motifs: RSX(pS/pT)XP and RXUX(pS/pT)XP, called mode I and II, respectively (X stands for any amino acid, and U stands for an aromatic or aliphatic amino acid) [33]. A recent study demonstrated that many other motifs conform to mode I when they have at least one basic residue in position −3 to −5 relative to the pS/pT site, and never a proline residue at position +1. Moreover, less than 50% of 14-3-3 mode I-related motifs have a proline at position +2 [34]. 

To find potential 14-3-3ζ/δ binding patterns in the sequence of PGAM2, we used the 14-3-3-Pred tool [35]. The analysis, which used the artificial neural network and support vector machine, revealed that the sequence 60LKRAIRTLWAI70 (Figure 5C) might recognize 14-3-3 proteins, with the following confidence indicators: artificial neural network value of 0.831 (cut-off = 0.55), and support vector machine value of 0.776 (cut-off = 0.25).

### 2.4. Regulation of Nuclear Import of PGAM2

In the present study, we demonstrated that the 14-3-3ζ/δ–PGAM2 interaction depends, at least indirectly, on the activity of the IGF1–PI3K signaling pathway (Figure 6C). 

The human 14-3-3ε and ζ/δ isoforms are known to regulate the activity of this pathway via interaction with insulin receptor substrate 1 and 2 (IRS1 and IRS2) and thus to modulate the sensitivity to insulin and IGF by interrupting the insulin/IGF receptor and IRS association [36]. Furthermore, the IRS1–14-3-3 interaction was shown to be completely inhibited by wortmannin [37], which emphasizes the role of PI3K in the regulation of this binding. PI3K itself is also an interaction partner for 14-3-3ζ/δ. It was shown that the ζ/δ isoform binds to the p85 regulatory subunit of PI3K and increases the translocation of the kinase to the plasma membrane, which is followed by an enhanced activation of AKT [38]. AKT, activated by PI3K via phosphoinositide-dependent kinase-1 (PDK1), directly inhibits the activity of GSK3β [39], and indirectly activates mTOR [40]. 14-3-3 proteins were demonstrated to regulate both GSK3β action via AKT [41,42] and mTOR via tuberous sclerosis protein 2 (TSC2, tuberin) [43]. TSC2, together with TSC1 (hamartin), inhibits RHEB, which is an inducer of mTOR, thus suppressing the activity of mTOR [44]. Upon the IGF1–PI3K pathway stimulation, AKT phosphorylates TSC2, which leads to a 14-3-3 protein binding to TSC2, which in turn reactivates (disinhibits) RHEB and activates mTOR [45]. Having found that the activity of the IGF1–PI3K signaling pathway regulates both the nuclear localization of PGAM2 (Figure 6A) and its interaction with 14-3-3ζ/δ (Figure 6C), we used inhibitors of AKT, GSK3β, and mTOR to determine their impact on the subcellular distribution of PGAM2. 

We show that the nuclear-to-cytoplasmic ratio of the fluorescence signal related to PGAM2 was unaffected when the cells were incubated for 2 h with the GSK3β inhibitor (SB216763), whereas the signal was significantly reduced in cells cultured with either the AKT or mTOR inhibitors (AKT Inhibitor IV and rapamycin, respectively) (Figure 7A). Specifically, the mean nuclear-to-cytoplasmic fluorescence intensity ratio was 0.85 ± 0.17 in the control experiment, 0.56 ± 0.1 in AKT inhibitor-treated cells, 0.92 ± 0.26 in GSK3β inhibitor-treated cells, and 0.63 ± 0.09 in mTOR inhibitor-treated cells (F = 31.96, *p* = 6 × 10^−16^, one-way ANOVA). This indicates that the nuclear localization of PGAM2 is regulated by the PI3K–AKT–mTOR but not PI3K–AKT–GSK3β pathway. The PI3K–AKT–mTOR pathway integrates signals from growth factors, nutrients and energy supply, and its activity promotes cell growth, metabolism, proliferation, and survival [46]. On the other hand, the inhibition of the pathway signaling, e.g., in hypoxic conditions, leads to autophagy [47]. Previously, the nuclear localization of PGAM2 has been linked with the maintenance of nucleolar structure, RNA synthesis, and assembly of new pre-ribosomal subunits [15]. Verification of whether the nuclear withdrawal of PGAM2 that is affected by inhibition of the IGF1–PI3K–AKT–mTOR pathway indeed resulted in the absence of PGAM2 in the nucleoli, we immunodetected PGAM2 C-end in propidium iodide counterstained cells. We found that in cells cultured in serum-free medium, and in the presence of PI3K or mTOR inhibitor, the nucleolar retention of PGAM2 was completely abolished (Figure 7B). However, 14-3-3ζ/δ is still present in the nucleoli under these conditions, which indicates that the 14-3-3ζ/δ–PGAM2 interaction is a prerequisite for PGAM2 nuclear import, but not export.

## 3. Materials and Methods

### 3.1. Protein Expression and Purification

Rabbit PGAM2 used in the crystallization experiments was obtained from the natural source – frozen rabbit skeletal muscle. For all other experiments, recombinant human PGAM2 expressed in *E. coli* cells was used. Wild-type (WT) human PGAM2 construct was obtained as expression-ready synthetic gene in pET-21a(+) plasmid from GenScript (Piscataway, NJ, USA). K49G, K129D and K146T mutations were introduced using GeneArt™ Site-Directed Mutagenesis System (ThermoFisher Scientific, Waltham, MA, USA) according to manufacturer’s protocol, using the following primers: PGAM2 p.K49G FW: GCCATCAAGGATGCCGGCATGGAGTTTGACATC, PGAM2 p.K129D FW: CCCGATGGACGAGGACCACCCCTACTACAAC, PGAM2 p.K146T FW: GGTACGCAGGACTCACACCCGGGGAACTCC. Respective reverse primers had exactly reversed complementary sequences.

Recombinant proteins were expressed overnight in HI-Control™ BL21(DE3) *E. coli* cells (Lucigen, Middleton, WI, USA) in LB-Broth at 37 °C. Pelleted cells were lysed using BugBuster Protein Extraction Reagent (Merck KGaA, Darmstadt, Germany). Rabbit skeletal muscle was homogenized mechanically in 50 mM TRIS-HCl pH 7.5 buffer at 4 °C, with 0.25 M sucrose, 1 mM EDTA, 1 mM PMSF and 1 mM β-mercaptoethanol. 

The following purification steps were the same for the natural rabbit and recombinant human proteins. Lysate was clarified by centrifugation and two-step salt fractionation was performed using 50% and 60% saturation of ammonium sulphate at 4 °C. Precipitate from the second step was resolubilized at 4 °C with 20 mM TRIS-HCl, pH 7.4, buffer with 0.1 mM EDTA and dialyzed against the same buffer. Finally, cellulose phosphate (Merck KGaA, Darmstadt, Germany) affinity chromatography column was applied, from which PGAM2 was specifically eluted with 10 mM 3-phosphoglyceric acid. PGAM2 yield and purity were controlled at each step using Bradford assay and by monitoring PGAM enzymatic activity as described previously [48].

### 3.2. PGAM2 Crystallization

Before crystallization, an additional size exclusion chromatography step was performed using HiLoad 16/600 Superdex 200 pg column (Cytiva, Marlborough, MA, USA), and the protein was concentrated to 6 mg/mL. Crystallization experiments were carried out at 292 K using the hanging-drop vapor-diffusion method in 3 µL drops mixed in 1:1 ratio from the protein and precipitant solutions. PGAM2 was crystallized using 0.1 M HEPES-NaOH, pH 7.0, 0.2 M NaCl and 20% PEG6000 as the precipitant solution.

### 3.3. X-ray Data Collection, Structure Solution and Refinement

Before diffraction experiments, the crystals were cryoprotected in their mother liquor supplemented with 20% (*v/v*) glycerol and then flash-vitrified at 100 K in a cold nitrogen-gas stream. For X-ray diffraction experiments, synchrotron radiation was utilized as provided by the beamline 14.2 of the BESSY II synchrotron, Berlin, Germany, equipped with a Rayonix MX-225 square CCD detector. The diffraction data were processed and scaled with XDSAPP [49,50]. The crystal structure was solved by molecular replacement using Phaser [51] and the PDB model 1YFK of human PGAM1 as the molecular probe. The initial model was manually rebuilt in Coot [52] and refined in several rounds of isotropic refinement in phenix.refine [53] alternating with manual adjustments in Coot. Ultimately, riding H atoms of the protein molecule were included in Fc calculations and the ADP (Atomic Displacement Parameter) model was changed to anisotropic, a move justified by the high resolution of the diffraction data. The R/R_free_ factors dropped on the transition from iso to aniso refinement from 0.186/0.210 to 0.152/0.186. Statistics of data collection, structure solution and refinement are presented in Table 1. The structure has been deposited in the PDB under the accession code 6H26. Raw diffraction images were deposited in the Macromolecular Xtallography Raw Data Repository (MX-RDR) with DOI 10.18150/repod.9330812.

### 3.4. Nucleolar Extract Isolation, PGAM2-Affinity Chromatography and Protein Identification

Nucleoli were purified from KLN-205 cells with the protocol described in [54] and treated with RNAse to extract nucleolar proteins. Recombinant human PGAM2 was immobilized on CNBr-activated Sepharose (Merck KGaA, Darmstadt, Germany) and packed into a glass gravity column. Next, the nucleolar protein extract was run through the resin three times. Non-specifically bound proteins were washed out with 20 mM TRIS-HCl buffer, pH 7.4, with 150 mM NaCl. Proteins interacting with PGAM2 were eluted using 20 mM TRIS-HCl buffer, pH 7.4, with 2 M NaCl. Protein identification was performed by LC-MS/MS at the Laboratory of Mass Spectrometry of the Institute of Biochemistry and Biophysics of the Polish Academy of Sciences, Warsaw.

### 3.5. Cell Culturing and Treatment

In vivo experiments were performed using KLN-205 cells (ATCC). Cultures were maintained at 37 °C under 5% CO2 in Eagle’s Minimal Essential Medium supplemented with L-glutamine (2 mM), 10% (*v/v*) non-essential amino acids, penicillin (100 units/mL), streptomycin (100 mg/mL), and 10% (*v/v*) fetal bovine serum. Prior to experimental procedures, the cells were cultured for 48 h under standard conditions. 

For the actual experiments, the standard medium was replaced by a serum-free medium for 2 h, or the standard medium was supplemented with the following inhibitors for 2 h: wortmannin (PI3K inhibitor, 2.3 µM), AKT inhibitor IV (AKT inhibitor, 1 µM), SB216763 (GSK3β inhibitor, 5 µM), and rapamycin (mTOR inhibitor, 0.01 µM).

### 3.6. Protein Labelling and Transfection

WT PGAM2 and its muteins were incubated overnight at 4 °C on rotation with fluorescein isothiocyanate (FITC). The molar ratio of the proteins to the fluorophore was 1:40. After incubation, unbound FITC particles were removed from the solution by a series of 2 min centrifugations using desalting columns (Cat. No. 89883, ThermoFischer Scientific, Waltham, MA, USA) at 2000× *g* and 4 °C. The efficiency of labelling was checked spectrophotometrically using Bradford assay for protein concentration and absorption at 495 nm for FITC. 

KLN-205 cells were transfected with labelled PGAM2 using ProteoJuice (Cat. No. 71281, Merck KGaA, Darmstadt, Germany). The transfection procedure was performed according to the manufacturer’s instruction using 3 µg of protein per each treated culture. 

### 3.7. Proximity Ligation Assay

In vivo protein interactions were detected using DuoLink In Situ Orange Starter Kit Mouse/Rabbit (Merck KGaA, Darmstadt, Germany) according to the protocol provided by the manufacturer. The following primary antibodies were used: rabbit anti-PGAM2 (1:1000, Cat. No. PA5-22038, ThermoFisher Scientific, Waltham, MA, USA), mouse anti-14-3-3ζ/δ (1:1000, Cat. No. ab188368, Abcam, Cambridge, UK), and mouse anti-RPLP0 (1:1000, Cat. No. ab88872, Abcam, Cambridge, UK). In control experiments, the primary antibodies were omitted.

### 3.8. Immunocytochemistry

Immunofluorescence studies were performed as described earlier [55]. After the experiment, the cells were fixed, permeabilized, blocked, and incubated overnight at 4 °C with the following primary antibodies: rabbit anti-PGAM2 (1:1000, Cat. No. PA5-22038, ThermoFisher Scientific, Waltham, MA, USA), goat anti-PGAM2 C-end (1:1000, produced and tested as described in [48]), mouse anti-14-3-3ζ/δ (1:1000, Cat. No. ab188368, Abcam, Cambridge, UK). The primary antibodies were detected using fluorophore-labelled secondary antibodies: goat anti-rabbit AlexaFluor 633 (1:2000, Cat. No. a21070, ThermoFisher Scientific, Waltham, MA, USA), goat anti-mouse AlexaFluor 488 (1:2000, Cat. No. ab150113, Abcam, Cambridge, UK), and anti-goat FITC (1:2000). In control experiments, the primary antibodies were omitted. 

After the fixation, the cells were counterstained with propidium iodide (0.2 µg/mL) for 2 min. Nuclei were counterstained with 4′,6-diamidyno-2-fenyloindol (DAPI) and the actin cytoskeleton was visualized with Phalloidin-ATTO488 or Phalloidin-ATTO633.

### 3.9. Confocal Microscopy, Fluorescence Measurement, and Statistical Analysis

Confocal microscopy analysis was performed as described earlier [55]. Quantification of the fluorescence signal was carried out using ImageJ software [56]. For the experiments with labelled PGAM2 transfection and in proximity ligation assays, the fluorescence signal was measured from the nuclei based on DAPI staining. In immunostaining experiments, the nuclear retention of PGAM2 was determined based on the ratio of nuclear PGAM2-related signal to the cytoplasmic signal. 

For comparison of more than two data sets, the one-way ANOVA with post hoc Tukey HSD test was used. *p*-value < 0.05 was considered to indicate significant difference. The results are expressed as the mean and standard deviation, and the data are visualized with box plots, wherein the center line shows the median, and × marks the mean. The bottom box limit indicates the 1st quartile, and the top box limit indicates the 3rd quartile. Whiskers range between the two extreme data points within a group. Outliers are placed beyond the whiskers. Data was analyzed using SigmaPlot11 (Systat Software, San Jose, CA, USA) and Microsoft Excel 2016 (Redmond, WA, USA) software. All the experiments were performed at least in triplicate.

## 4. Conclusions

In this report, we have presented the first experimental 3D structure of the muscle isoenzyme, PGAM2, of phosphoglycerate mutase. Based on the structural data, we have identified and experimentally confirmed the functionality of residues, such as K146, involved in nuclear import of PGAM2. The full NLS epitope can be formed only upon dimerization of PGAM2, which also explains why the PGAM2/PGAM1 heterodimers (but not the PGAM1/PGAM1 homodimers) of the two isoforms undergo translocation to the nucleus. We have also presented several lines of evidence confirming that PGAM2 interacts with 14-3-3ζ/δ in vivo and that nucleolar localization of PGAM2 depends on the activity of the IGF–PI3K–AKT–mTOR pathway, as summarized in Figure 8.

## Figures and Tables

**Figure 1 ijms-23-13198-f001:**
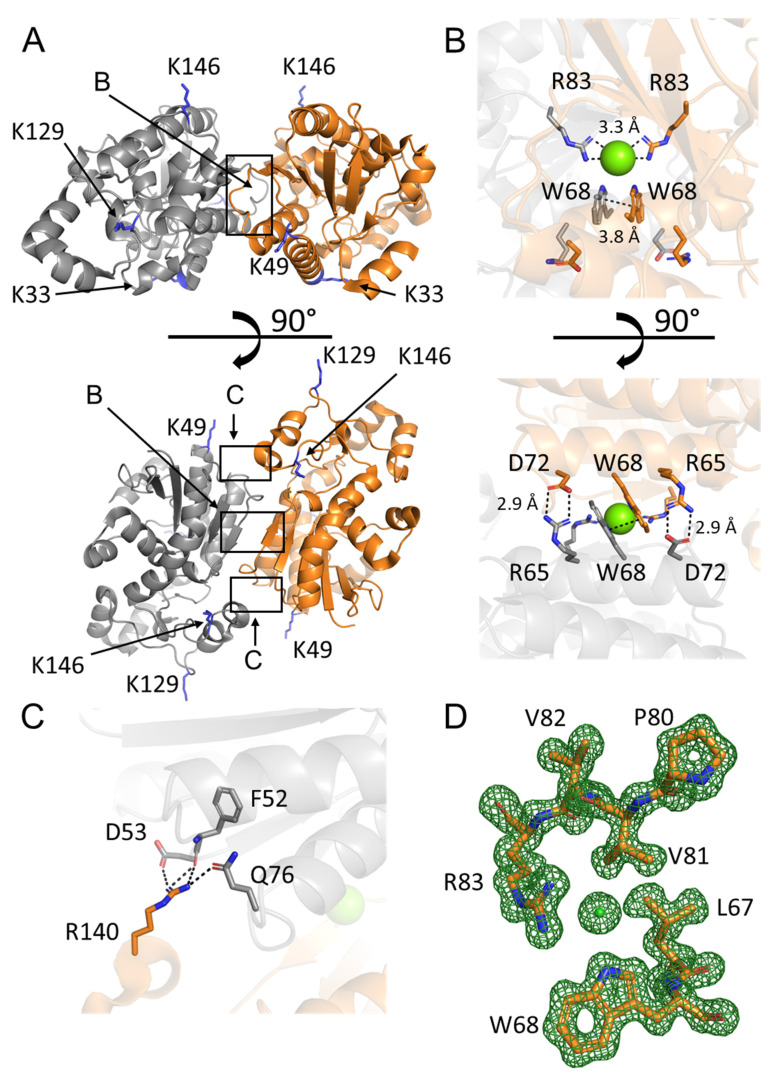
Structure of rabbit muscle PGAM2. (**A**) Overview of the PGAM2 structure: location of close-up views is indicated with labeled arrows; orange, gray—subunits of the PGAM2 homodimer, blue—surface exposed positively charged residues present in PGAM2 but not in PGAM1 (K33, K49, K129, K146). (**B**) Interactions in the central hydrophobic region 60–83; green—chloride ion. (**C**) Interactions at the edge of subunit interface. (**D**) Omit Fo-Fc electron density map of the residues surrounding the chloride ion at the subunit interface, rendered at 3σ.

**Figure 2 ijms-23-13198-f002:**
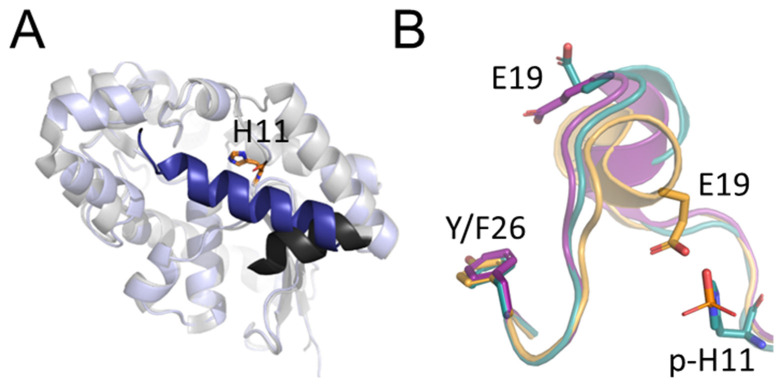
Structure of PGAM2 compared with PGAM1 and Alpha Fold prediction. (**A**) Superposition of rabbit PGAM2 crystal structure (gray) and Alpha Fold prediction of human PGAM2 structure (blue). C-terminal region in disagreement with prediction is highlighted in darker colors. Catalytic H11 residue in orange. (**B**) Difference in position of E19 in PGAM2 (purple), dephosphorylated PGAM1 (orange, PDB ID 4GPI) and phosphorylated PGAM1 (cyan, PDB ID 4GPZ).

**Figure 3 ijms-23-13198-f003:**
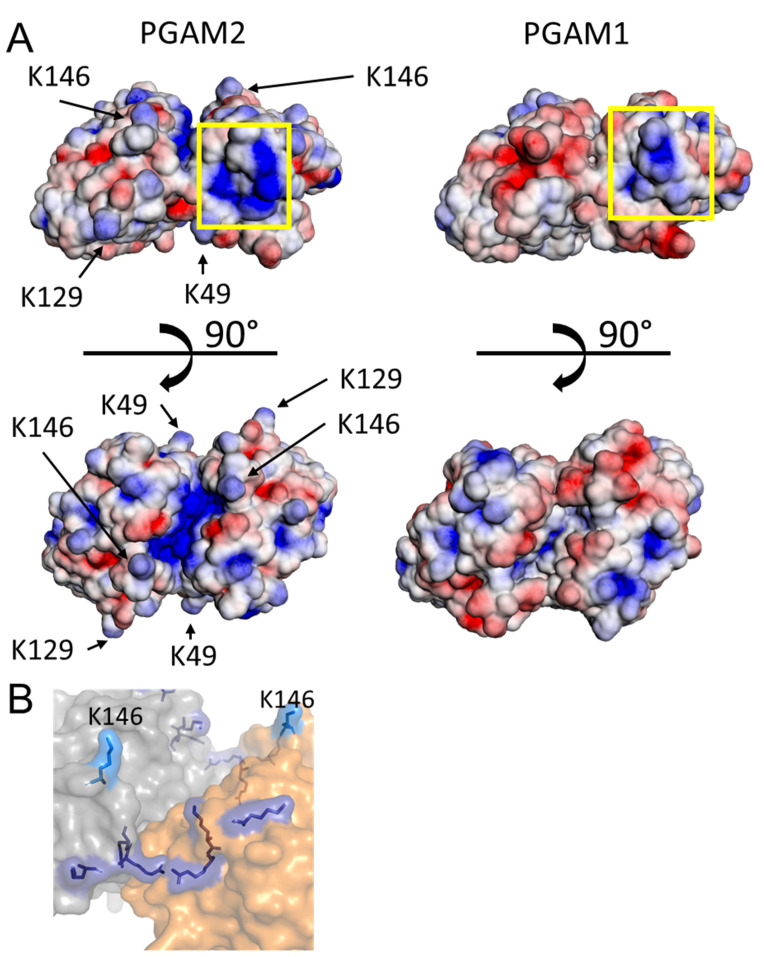
Lysine residues potentially engaged in nuclear import of PGAM2. (**A**) Solvent accessible surface potential of PGAM1 (PDB ID 4GPI) and PGAM2, colored from −3 kT/e (red) to 3 kT/e (blue). Calculated by APBS [20]. Yellow box—positive surface patch common to both PGAM1 and PGAM2. (**B**) Proposed NLS of PGAM2: orange, gray—subunits of the PGAM2 homodimer, light blue—K146 residue necessary for proper nuclear import, present in PGAM2 but not PGAM1, dark blue—positively charged residues in the vicinity of K146, conserved in both PGAM1 and PGAM2 (K138, R140, R141, K176, K179, R180).

**Figure 4 ijms-23-13198-f004:**
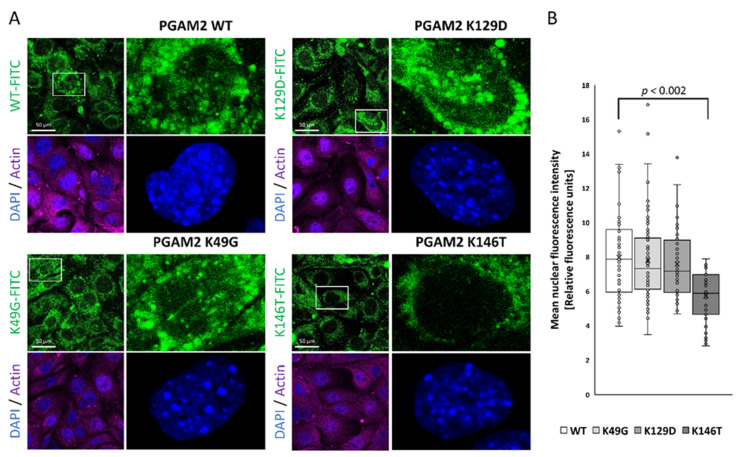
The nuclear import of PGAM2 depends on the lysine residue 146. (**A**) The fluorescence signal visible in the regions of the nuclei recorded for fluorophore (FITC)-labelled PGAM2 muteins K49G and K129D did not differ from that observed in WT-transfected cells. Only the mutein K146T-related fluorescence was significantly lower than that observed in WT-transfected cells. (**B**) Quantification of labelled PGAM2-related fluorescence from the regions of the nuclei.

**Figure 5 ijms-23-13198-f005:**
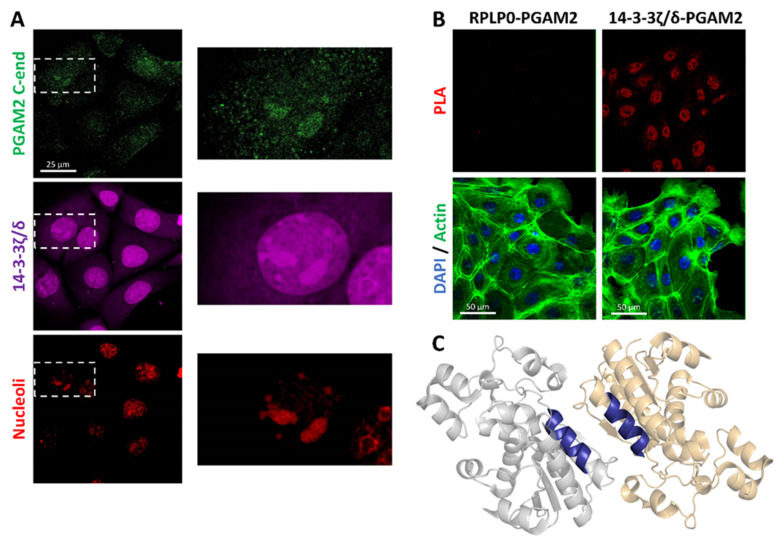
In vivo detection of PGAM2 interaction with RPLP0 and 14-3-3ζ/δ. (**A**) PGAM2 co-localizes with 14-3-3ζ/δ in the nucleoli, as determined using propidium iodide counterstaining. (**B**) Proximity ligation assay (PLA) did not show the presence of RPLP0–PGMA2 interaction within the cells. On the other hand, 14-3-3ζ/δ–PGAM2 complexes were observed, mainly in the cellular nuclei. (**C**) Hypothetical 14-3-3–binding site on the surface of PGAM2 dimer (gray and orange, PGAM2 subunits; blue, PGAM2 sequence: 60LKRAIRTLWAI70 which may recognize 14-3-3 proteins.

**Figure 6 ijms-23-13198-f006:**
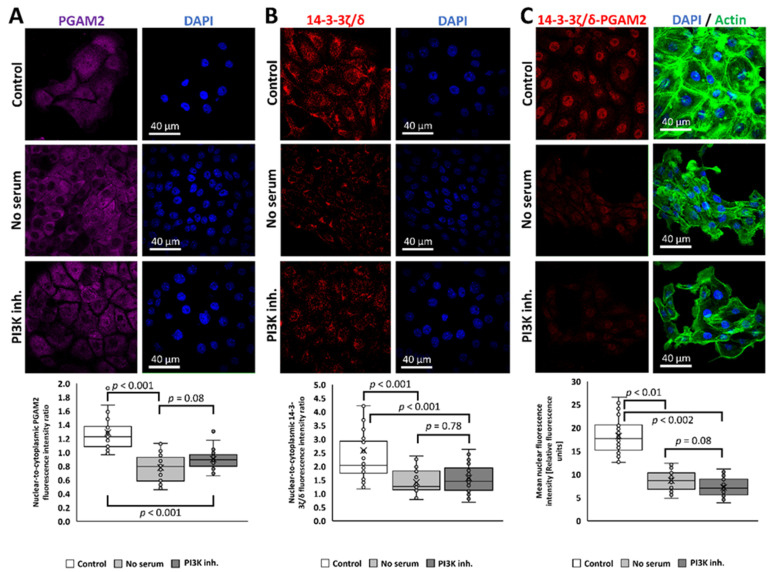
14-3-3ζ/δ–PGAM2 interaction depends on the insulin/IGF1–PI3K signaling pathway. PGAM2 (**A**) and 14-3-3ζ/δ (**B**) are not present in the nucleus when cells are cultured in serum-depleted medium or in the presence of PI3K inhibitor, wortmannin. Similarly, the interaction between these proteins (**C**) is almost completely abolished in the above conditions.

**Figure 7 ijms-23-13198-f007:**
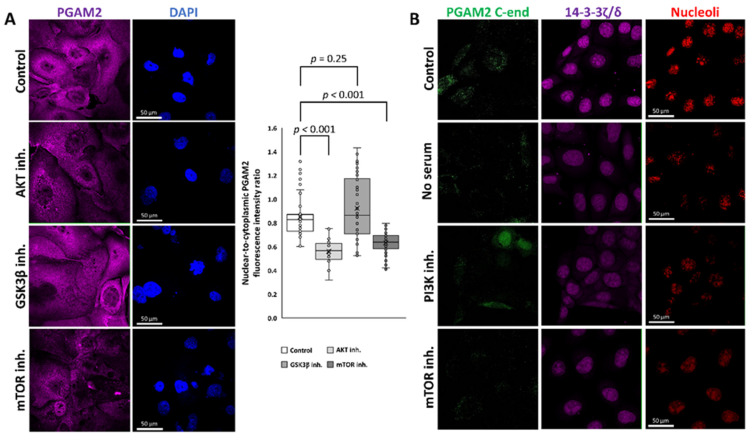
Regulation of nuclear localization of PGAM2. Immunodetection of PGAM2 showed that in the presence of Akt and mTOR inhibitors the protein nuclear retention is significantly reduced in comparison to untreated cells. In contrast, the inhibition of GSK3β activity did not decrease the amount of nuclear PGAM2 (**A**). Additionally, the inhibition of the IGF1–PI3K–AKT–mTOR pathway abolished the nucleolar localization of PGAM2 (**B**).

**Figure 8 ijms-23-13198-f008:**
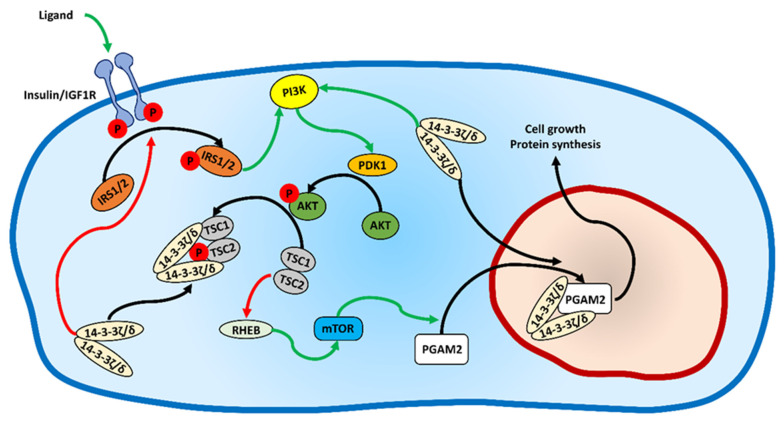
Schematic representation of the insulin/IGF1R–PI3K–AKT–mTOR signaling pathway and its impact on the subcellular distribution of PGAM2. The regulatory function of 14-3-3ζ/δ on the pathway is also included in the scheme. Arrows: green—activation; red—inhibition; black—translocation/targeting.

**Table 1 ijms-23-13198-t001:** Details of X-ray data collection and refinement. Numbers in parentheses refer to the highest resolution shell.

Data Collection	
Radiation source	BESSY II, Berlin
Beamline	14.2
Wavelength [Å]	0.9184
Temperature [K]	100
Space group	*P*4_1_2_1_2
Unit-cell dimensions [Å]	*a* = *b* = 71.308, *c* = 112.48
Resolution [Å]	33.99–1.288 (1.37–1.288)
Total/unique reflections	1386601/73677
Completeness (%)	99.7 (98.6)
Multiplicity	18.8 (16.7)
R_merge_	0.057 (1.155)
<I/σ(I)>	31.05 (2.88)
CC_1/2_	0.999 (0.749)
Wilson B-factor [Å^2^]	14.6
**Refinement**	
Refinement resolution [Å]	33.99–1.29
R_work_/R_free_	0.152/0.186
Unique/free reflections	73675/1011
Matthews coefficient [Å^3^/Da]	2.49
Solvent content (%)	50.6
No. of non-H atoms/<B> [Å^2^]:	
Protein	1948/20.2
Solvent	156/26.5
Heterogen	9/27.6
RMSD bond lengths [Å]	0.01
RMSD bond angles [°]	1.07
Clashscore	1.94
Ramachandran favored (%)	97.1
Ramachandran allowed (%)	2.9
PDB ID	6H26

## Data Availability

Atomic coordinates and structure factors have been deposited in the Protein Data Bank (PDB) under accession code 6H26. Raw diffraction images were deposited in the Macromolecular Xtallography Raw Data Repository (MX-RDR) with DOI 10.18150/repod.9330812.

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
