# Peer review of "High-Resolution Crystal Structure of Muscle Phosphoglycerate Mutase Provides Insight into Its Nuclear Import and Role"

_ijms, 2022, doi:10.3390/ijms232113198_

Round 1

Reviewer 1 Report

The manuscript “High-resolution crystal structure of muscle phosphoglycerate mutase provides insight into its nuclear import and role” first provides the structure of PGAM2 and uses the cell and fluorescence measurement checking by confocal microscopy shows the nuclear localization of the protein and also shows the interaction with 14-3-3ζ/δ and novelty and important. However, it still has some minor issues in the manuscript.

  1. In the introduction part, the author needs to introduce what “14-3-3ζ/δ” is. Are there any reports about this regulatory signaling molecule interaction with PGAM2?
  2. In Fig.1, the author needs to list the distance between the interaction atom to make sure it is the salt interaction. How does the author know the chloride ion instead of water?
  3. In Fig1A, label all K33, K49, K129, K146 but not only K146
  4. The PDB for PGAM1 is 4GPZ, not 4PGZ.
  5. The figure needs to be sequenced in the manuscript. As in the manuscript, Fig. 3D needs to combine with Fig.2.
  6. Fig.2B is confusing. PGAM2, dephosphorylated PGAM1 and phosphorylated PGAM1 are better shown in different colors, and what are the PDBs of dephosphorylated PGAM1 and phosphorylated PGAM1? It does not show Y26, and the stick structure of H11 is also wrong.

Author Response

Thank you very much for your comments and suggestions. We improved the manuscript accordingly. Below, the reviewer's comments are listed in bold, and the authors' responses are in italics.

In the introduction part, the author needs to introduce what “14-3-3ζ/δ” is. Are there any reports about this regulatory signaling molecule interaction with PGAM2?

To the best of our knowledge there is no report regarding the interaction between 14-3-3 protein and PGAM2. According to the suggestion of the reviewer we added a paragraph in the introduction section describing 14-3-3 proteins (lines 57-66, highlighted in yellow).

In Fig.1, the author needs to list the distance between the interaction atom to make sure it is the salt interaction. How does the author know the chloride ion instead of water?

Distance was added to Figure 1. The distance is consisted with chloride-arginine distance most commonly observed in structures in PDB (within 3.4 A). We are quite certain of the presence of chloride as inserting water in this position results in a large positive peak in the Fo-Fc electron density map and chloride ions are present in analogous positions in published PGAM1 structures.

In Fig1A, label all K33, K49, K129, K146 but not only K146

Labels for K33, K49 and K129 were added.

The PDB for PGAM1 is 4GPZ, not 4PGZ.

Mistake was corrected 4PGZ was changed to 4GPZ.

The figure needs to be sequenced in the manuscript. As in the manuscript, Fig. 3D needs to combine with Fig.2.

Sequence of the figures was changed. Figure 3D was expanded and transferred to Supplementary Figure 1. Figures 3 A,B,C were modified and merged into Figure 3A. Figure 2C was moved to Figure 3B. References in the manuscript text were changed accordingly.

Fig.2B is confusing. PGAM2, dephosphorylated PGAM1 and phosphorylated PGAM1 are better shown in different colors, and what are the PDBs of dephosphorylated PGAM1 and phosphorylated PGAM1? It does not show Y26, and the stick structure of H11 is also wrong.

Figure 2B was edited for clarity. Additionally, a clearer explanation of the influence of Y26 phosphorylation on PGAM1 activity was given, based on the results described in Reference [2].

Reviewer 2 Report

In this manuscript Wiśniewski et al. report to have solved the 3D structure of phosphoglycerate mutase 2 (PGAM2) , a glycolytic enzyme. The authors claim to have identify the residues involved in the nuclear localisation of the protein and potentiel interaction partners in the nucleolus.

The manuscript is very well written with very few mistapping errors. I do, however, have some comments that would require consideration, mainly regarding the structural biology part of the paper.

1) Figures 1 to 3 are difficult to read and should be improved for better clarity. In Fig. 1, removing of the molecular surface and shadows, adding depth cueing and showing only the sidechains (when possible) should give a better picture of the dimer. Please also label all lysine residues in blue. For panel 1D, it would be better to show a composite omit map around the residues you want to highlight as 2fo-fc maps are bias by the built model. In panel 2B, the conformational change is really difficult to see with this type of representation. Please also show the sidechain of residue 26 as it is involve in the repositionning. In Fig. 3, a general view of the molecular surface of the dimer (colored by electrostatic potential) would be more informative than close-up views on the potential lysine residues involved in nuclear import. In the same figure, another sequence alignment with more species would show which lysine residues are strictly conserved in PGAM2 and could be important for the presence of PGAM2 in the nucleolus.

2) K33 was not mutated to look for important residues involved in the NLS because it is close to the active site of the enzyme but indeed the catalytic activity of PGAM2 and its presence in the nucleolus may not be related. Do the authors have further evidence that this residue is not important for the nucleolar localisation of PGAM2?   

3) PGAM2 is subject to several post-translational modifications, but none are visible in the high resolution crystal structure whereas the protein was extracted from a natural source. What might be the explanation? Do the purification steps get ride of some modified PGAM2?

4) Looking at the list (Supplementary Table 1) of identified proteins able to bind to PGAM2, it seems to me that there is no obvious canditate for the transportation of the PGAM2 cargo into the nucleus. Furthermore, the authors make no mention in the core of the manuscript of a putative transport protein that could recognise the putative quaternary NLS of PGAM2. Could the presence of PGAM2 in the nucleolus be due solely to its pI (8.99) and its interactions with a specific RNA/DNA structure, whereas PGAM1 is an acidic protein (pI=6.67) and may not be able to bind any RNA/DNA?   

Typos:

Page 4, Table 1 – Put values of refinement resolution and not cell parameters.

Page 6, line 107 – “gives an RMSD”.

Page 7, line 175 – K146 is not in bold in the sequence.

Author Response

Thank you very much for the suggestions. We improved the manuscript accordingly. Below, the reviewer's suggestions are listed in bold, and the authors' responses are in italics.

1) Figures 1 to 3 are difficult to read and should be improved for better clarity. In Fig. 1, removing of the molecular surface and shadows, adding depth cueing and showing only the sidechains (when possible) should give a better picture of the dimer. Please also label all lysine residues in blue. For panel 1D, it would be better to show a composite omit map around the residues you want to highlight as 2fo-fc maps are bias by the built model. In panel 2B, the conformational change is really difficult to see with this type of representation. Please also show the sidechain of residue 26 as it is involve in the repositionning. In Fig. 3, a general view of the molecular surface of the dimer (colored by electrostatic potential) would be more informative than close-up views on the potential lysine residues involved in nuclear import. In the same figure, another sequence alignment with more species would show which lysine residues are strictly conserved in PGAM2 and could be important for the presence of PGAM2 in the nucleolus.

Figure 1-3 were edited for clarity. Map in Figure 1D changed to omit map. Figure 3A,B,C were modified and merged into Figure 3A, showing the surface potential of the whole dimer, as suggested by the reviewer. Additional mammalian species were added to alignment of PGAM1 and PGAM2. The alignment was moved to Supplementary Figure 1 because of space constraints.

2) K33 was not mutated to look for important residues involved in the NLS because it is close to the active site of the enzyme but indeed the catalytic activity of PGAM2 and its presence in the nucleolus may not be related. Do the authors have further evidence that this residue is not important for the nucleolar localisation of PGAM2?

We do not have evidence that K33 is not involved in nuclear localization of PGAM2, it might as well be. However, we decided to err on the side of caution and exclude K33 form our mutational studies to avoid any artifacts that may have been introduced by altering the important catalytic function of PGAM2.

3) PGAM2 is subject to several post-translational modifications, but none are visible in the high resolution crystal structure whereas the protein was extracted from a natural source. What might be the explanation? Do the purification steps get ride of some modified PGAM2?

We observed residual density that may indicate n-terminal acetylation, however we do not see any evidence of phosphorylation. Removal of phosphorylation during purification is plausible, as no phosphatase inhibitors were used. Moreover, one of the purification steps involved binding of the protein to negatively charged phosphocellulose resin which may enrich the preparation in dephospho-PGAM.

4) Looking at the list (Supplementary Table 1) of identified proteins able to bind to PGAM2, it seems to me that there is no obvious canditate for the transportation of the PGAM2 cargo into the nucleus. Furthermore, the authors make no mention in the core of the manuscript of a putative transport protein that could recognise the putative quaternary NLS of PGAM2. Could the presence of PGAM2 in the nucleolus be due solely to its pI (8.99) and its interactions with a specific RNA/DNA structure, whereas PGAM1 is an acidic protein (pI=6.67) and may not be able to bind any RNA/DNA?

Regarding the PGAM2 transport into a cell nucleus, the interactome analysis revealed that importin beta may be one of the PGAM2-interacting proteins (line 45 in the supplementary table). Although importin beta does not bind directly to nuclear localization signal of the cargo protein, it suggests that the importin system may be responsible for PGAM2 nuclear import.  

On the other hand, it cannot be excluded that PGAM dimers (homodimers PGAM2-PGAM2 and heterodimer PGAM2-PGAM1) diffuse freely through nuclear pore which is permeable for molecules of the mass below about 50 kDa (the mass of PGAM dimer is about 54 kDa). And just because of positive charge of the dimers containing PGAM2 such a complex may be retained in nuclei. 

However, to the best of our knowledge, the isoelectric points for PGAM isoforms were calculated for the unfolded PGAMs only, thus more detailed experimental studies on the real pI for PGAMs in their tertiary and quaternary conformation are needed to verify this intriguing hypothesis.

Additionally, all the typos have been corrected.